# Multi-Scenario Simulation of Land Use and Habitat Quality in the Guanzhong Plain Urban Agglomeration, China

**DOI:** 10.3390/ijerph19148703

**Published:** 2022-07-17

**Authors:** Hao Ye, Yongyong Song, Dongqian Xue

**Affiliations:** School of Geography and Tourism, Shaanxi Normal University, Xi’an 710119, China; yehao7474@163.com

**Keywords:** land-use change, habitat quality, InVEST model, scenario simulation, Guanzhong Plain urban agglomeration

## Abstract

Regional habitat quality is a proxy of biodiversity. Simulating changes in land use and habitat quality in urban agglomerations is the scientific basis for promoting the optimal allocation of land resources and building ecological civilizations in urban agglomerations. Therefore, we established a research framework mainly consisting of the Future Land Use Simulation (FLUS) model with the Integrated Valuation of Environmental Services and Tradeoffs (InVEST) model to predict the spatial and temporal distribution of habitat quality. In addition, we set three scenarios which were a natural development scenario, a cultivated land protection scenario, and an ecological protection scenario to analyze the changes of habitat quality in the Guanzhong Plain urban agglomeration in 2035. The results showed that: (1) the FLUS model had an excellent effect on the simulation of land-use change in the Guanzhong Plain urban agglomeration, with an overall accuracy of 0.952 and a kappa coefficient of 0.924. (2) From 2000 to 2035, the cultivated land area of the study area, which was mainly transferred into construction land and grassland, shrank due to the process of urbanization. (3) The habitat quality score of this region gradually decreased from 2000 to 2020, and it continued to decrease to 0.6921 in 2035 under the natural development scenario, while it increased under the other two scenarios. The low-value areas of habitat quality were mainly located in the middle of this region with Xi’an as the core, whereas the high-value areas were mainly distributed in the southern Qinling Mountains and the northern Loess Plateau. (4) Of the different scenarios, the ecological protection scenario had the highest habitat quality, while the natural development scenario had the lowest. Besides this, we also found that the cultivated protection scenario had high habitat quality, which was mainly because the rate of occupation of ecological land was controlled. The results are expected to provide a scientific basis for optimizing the spatial allocation of land resources and promoting the sustainable use of land space in other ecologically fragile urban agglomerations.

## 1. Introduction

Since the 1950s, with the continuous and rapid advancement of urbanization and industrialization in the world, the intensity and breadth of human activities on the surface of natural ecosystems have reached an unprecedented level, resulting in the loss of biodiversity, habitat fragmentation, and ecosystem degradation [1,2,3], which seriously threaten human well-being and ecosystem stability [4,5]. Especially in urbanized areas, human activities and urban spatial expansion have caused a dramatic transition in land-use patterns, which change the structure and composition of habitats, which aggravates habitat fragmentation and hinders the process of material exchange and energy flow between habitat patches [6,7]. Habitat quality, an important indicator of biodiversity, refers to the ability of ecosystems to provide the necessary production conditions for individuals or populations and reflects the ecosystem health and the ecological balance status [8]. Therefore, scientific simulation of land-use changes in urbanized areas and their impact on habitat quality is urgently needed to formulate regional land-use policies and protect the ecological environment for the United Nations sustainable development goals 2 and 15 [9,10].

In recent years, scholars at home and abroad have conducted extensive research on land use and habitat quality from different perspectives, using different temporal and spatial scales and methods. Land-use change is not only an important driving force for the evolution of habitat quality, but it also profoundly affects the structure and function of regional ecosystems [11]. Zhang et al. [12] analyzed the impact of land-use change on habitat quality in the Pan Yangtze River Delta and found that the expansion of urban, built-up areas is an important factor causing the fragmentation of habitat patches. Mengist et al. [13] studied the impact of Land-Use/Cover Changes (LUCC) on habitat quality in the Kafa BR, Ethiopia, and found that the expansion of man-made landscapes (such as settlements and agricultural land) is the key factor affecting habitat quality in the reserve. Generally, based on field surveys and statistical data, scholars in early research evaluated the impact of land-use change on the habitat quality of a single species by constructing multiple evaluation index systems [14]. At present, the Integrated Valuation of Ecosystem Services and Tradeoffs (InVEST) model, jointly developed by Stanford University, the Nature Conservancy, and the World Wide Fund for Nature, overcomes the non-spatial approach to evaluating ESs, using purely statistical approaches that are valid independent of the characteristics of the space. The InVEST model has been widely used in regional habitat quality assessment. It can analyze the impact of different threat factors and land-use changes on regional biodiversity on multiple scales and can realize the visual expression of assessment results [15,16]. In terms of habitat quality assessment, Zhang et al. [17] and Fan et al. [18] used the InVEST model to study the temporal and spatial variation characteristics of habitat quality in Wenzhou and Liulin County, China, respectively. Similarly, Anseyee et al. [19] and Fan et al. [20] evaluated the correlations between habitat quality and influencing factors in the Winike Watershed and Hung River Valley, respectively. However, extant research assessing the impact of land-use changes on habitat quality in small- and medium-sized regions and habitat quality in large-scale regions [21] is relatively weak.

An important contributor to the overall development of China’s “the Belt and Road”, the construction pattern of ecological civilization, and the strategic layout of new urbanization, the Guanzhong Plain urban agglomeration is the core area in which to promote ecological protection and high-quality development in the middle and upper reaches of the Yellow River, China [22,23]. With the continuous expansion of urban agglomeration and steady progress of infrastructure construction, the contradiction between regional urbanization and ecological environment protection has gradually intensified [24,25,26]. Considering the background of the implementation of the national, new urbanization strategy, ecological protection and the high-quality development strategy of the Yellow River Basin, it is vital to improve the scientific understanding of the trend of land use in the Guanzhong Plain urban agglomeration on habitat quality and simulate the evolution trend of land use and habitat quality. These advances can be used to build a sustainable land space protection pattern and realize regional, high-quality development to achieve China’s plan for achieving socialist modernization by 2035. Therefore, based on land-use data for the Guanzhong Plain urban agglomeration in 2000, 2010, and 2020, we coupled the FLUS model with the InVEST model to simulate the regional land-use change pattern in 2035 under three scenarios, which are a natural development scenario, a cultivated land protection scenario, and an ecological protection scenario. Then, we evaluated the corresponding regional habitat quality response patterns. This study takes the Guanzhong Plain urban agglomeration as the study area, which can not only provide a scientific basis for optimizing the spatial allocation of land resources in other ecologically fragile urban agglomerations, but also provide a reference for promoting the sustainable use of land space.

## 2. Literature Review and Research Framework

### 2.1. Literature Review

Land-use/land-cover change is not only an environmental issue in the process of regional development, but also one of the most important environmental issues of global concern [27,28]. Since the 1980s, global change research has begun to focus on the impact of human activities on the natural environment, and its research goal is mainly to explore the impact of land-use change on global environmental change [29,30]. In 1995, the International Geosphere Biosphere Project (IGBP) and the International Human Dimension Project (IHDP) proposed the LUCC research program, which aims to study the impact of human systems on ecosystems through land-use change and its relationship with global change [31]. Subsequently, in 2005, these two projects launched the Global Land Project (GLP) [32] to explore how changes in ecosystem service functions affect human well-being [33]. Land-use/land-cover change is mainly driven by the interaction of political, economic, and land management systems and other factors. Urbanization, as the core driving force that promotes land-use change, has brought many benefits to regional development, such as economic growth, infrastructure improvement, and more employment opportunities [34,35], but it has also produced a series of problems, such as the disorderly expansion of construction land, deforestation, and farmland loss [36]. Existing research conclusions show that changes in land-use patterns have led to biodiversity loss, ecosystem disintegration, and a reduction in the connectivity of biological populations [37,38]. 

Land use dynamic simulation is the focus of current research on land-use change. Combining land-use models with scenario development can provide more explicit spatial predictions, which are of great significance for land-use management and policy making [39]. Common land-use prediction models include the System Dynamics Model [40], CLUE-S [41], SLEUTH [42], cellular automata, and the Markov model. However, existing studies show that it is difficult to reflect the changes in the characteristics of a land-use spatial pattern using the SD model due to a lack of processing function for spatial factors. Although the Markov model can predict the quantitative change of land use well, it is difficult to reflect the spatial change of land use using this model. Meanwhile, the CA model can only define transformation rules from the interaction between local individuals and has limitations in simulating the complexity of the evolution of geographical entities [43]. With technological progress and model improvement, the Geographical Simulation and Optimization System (GeoSOS), which includes CA, a Multi-Agent System (MAS), and Swarm Intelligence (SI), is beginning to be applied to land-use change simulation. In addition, the Future Land Use Simulation (FLUS) model, which is improved by GeoSOS [44], is effective in simulating multi-scale land-use change. This model not only solves the complex problems relating to the transformation rules and parameter determination of CA, but also introduces an adaptive inertia mechanism to better simulate the land-use change driven by human activities. With the help of this model, Chu et al. [45] simulated the spatial pattern change of the Beijing–Tianjin–Hebei urban agglomeration in 2030 and evaluated its impact on green space fragmentation, finding that urban expansion exacerbated the fragmentation of farmland and grassland. By coupling the FLUS model with the InVEST model, Ding et al. [46] simulated the habitat quality changes of Dongying City in 2030 under four scenarios: business as usual, fast cultivated land expansion, ecological security, and sustainable development. They found that the sustainable development scenario had the highest habitat quality, and it was most conducive to the coordinated development of urban development, agricultural production, and ecological protection. 

As an indicator of ecosystem health, habitat quality determines the ability of the ecosystem to provide services and commodities [47]. In recent years, scholars have begun to pay attention to the evaluation and analysis of habitat quality, especially the impact of land-use change on habitat quality. On the one hand, by using the ecological networks evaluation method [48], the habitat suitability index model [49], or constructing a multiple evaluation index system [50], some scholars evaluated habitat quality from the perspective of ecosystem attributes. On the other hand, by using a mathematical optimization algorithm [51] and MAXENT model [52], some scholars discussed the threat of land-cover change to biodiversity from the perspective of human social and economic activities. At present, in order to adapt to the complexity of the ecosystem and the differences in researchers’ understanding, the InVEST model is applied to multi-scale research and analysis to meet multiple service functions and multiple goals. In addition, it is used to assess the health of ecosystems [53,54].

### 2.2. Research Framework

Based on the land-use data for the Guanzhong Plain urban agglomeration in 2000, 2010, and 2020 and data on natural and social driving factors, we used the Markov model to predict the quantity and scale of multi-scenario land use in 2035. Taking this as the data basis, the FLUS model was used to simulate the change in the land-use spatial pattern under three scenarios. Based on the land-use data obtained for each scenario in 2035, we evaluated the temporal and spatial evolution characteristics of habitat quality under different scenarios using the InVEST model. The overall research framework is illustrated in Figure 1.

In this paper, the Markov model is based on the random process principle that the state at any moment in a finite time series is only related to the state at the previous moment, as proposed by the mathematician Markov [55]. By calculating the transition probability between different states, it can determine the trend of different land-use types over time and realize the prediction of land-use change. The FLUS model is a comprehensive model that combines human and natural driving forces to simulate multiple types of land-use scenario [44]. It is composed of an Artificial Neural Network (ANN) algorithm and a cellular automata module based on an adaptive inertia mechanism. Compared with traditional cellular automata, it can better simulate land-use change under the influence of human activities and natural conditions. The model firstly uses an Artificial Neural Network (ANN) algorithm to calculate the data of land use and various driving factors in the base period so as to obtain the adaptability probability of each land type. Secondly, it combines the adaptive probability with domain factors, the adaptive inertia coefficient, and the cost matrix. Finally, according to the roulette competition mechanism, it deals with the complexity and uncertainty of land-use change under the influence of various driving factors and obtains the simulation results of land-use change. Although the Markov model is good at predicting changes in the amount of land, it has some limitations in dealing with the spatial change of land use. Therefore, coupling the Markov model with the FLUS model can predict changes in land-use quantity and space [56]. The habitat quality module of the InVEST model mainly evaluates the habitat distribution and degradation under different scenarios by establishing the relationship between land-use status and threat sources. The research results reflect the response trend of habitat quality in relation to the change of land-use spatial pattern.

## 3. Materials and Methods

### 3.1. The Study Area

The Guanzhong Plain urban agglomeration is located at 104°34′–112°34′ E, 33°34′–36°56′ N, connecting the Qinling Mountains in the south, the Yellow River in the east, and the Loess Plateau in the north. It is the core for the development of the western region and the strategic fulcrum that connects the eastern and western regions of China. 

This region consists of Shaanxi, Shanxi, and Gansu Provinces, containing 11 prefecture-level cities, with a total land area of 1.07 × 10^5^ km^2^ (Figure 2). As the largest urban agglomeration in Northwest China, this area is a key area for the comprehensive management of environmental pollution and ecological protection. The region has a long history of development, intense human activity, and fragile ecological environments. The land in this region is mainly used for agricultural production, construction of transportation facilities, optimization of modern industrial structures, etc. The habitat types mainly include forest, grassland, river, and cultivated land. Now, the proportion of woodland and grassland area is 21.74% and 27.41%, respectively. Due to urban development and construction, the area of cultivated land is gradually decreasing, and water pollution is serious. In 2019, the resident population of the entire agglomeration was 44.85 million, the regional gross domestic product was CNY 216 million, and the proportion of regional construction space in the total area of the region was 5.73%, which is far higher than the limit of land development intensity (4.62%) stipulated in the National Land Planning Outline (2016–2030). Since the implementation of the western development strategy, with the rapid advancement of urbanization and industrialization, the regional land-use pattern has been drastically reconstructed, and the disorderly expansion of urban space has occupied cultivated land and ecological land, resulting in habitat fragmentation and the decline of ecosystem service function. Therefore, regional habitat quality urgently needs to be optimized and improved. In the context of implementing the new urbanization and ecological civilization strategies, it is of great significance to simulate the regional land-use change and the evolution trend of habitat quality under different development scenarios.

### 3.2. Data Sources

Land-use data in this study were obtained from the Resource and Environmental Science and Data Center of the Chinese Academy of Sciences (http://www.resdc.cn accessed on 6 April 2022) with a spatial resolution of 30 m × 30 m. They were subdivided into six first-class land types according to the National Land Use Classification System: cultivated land, woodland, grassland, waters, construction land, and unused land (Table A1), with a classification accuracy of more than 94%. The land-use drivers (Figure A1) used in this study for the FLUS model included physical (digital elevation model, slope), social (gross domestic product density, population density), and accessibility factors (distance from towns, rivers, national, provincial, and railways). Slope data were extracted from Digital Elevation Model (DEM) data downloaded from the Geospatial Data Cloud (http://www.gscloud.cn accessed on 10 April 2022); gross domestic product and population data were from the statistical yearbooks of Shaanxi Province, Gansu Province, and Shanxi Province; urban and river data were extracted from land-use type data; administrative boundary and traffic network data were from the National Basic Geographic Information Center (http://ngcc.sbsm.gov.cn accessed on 10 April 2022); and the distances from towns, rivers, and national roads were calculated by using distance analysis in ArcGIS software (Esri, Redlands, CA, USA). 

### 3.3. Methodology

#### 3.3.1. Land-Use Change Scenario Settings

Overall regional planning dominates the land supply at the macro level to meet different development demands, while different development needs create different land spatial patterns. Therefore, setting different scenarios to predict and simulate land-use change is of great significance for decision-makers to allocate land resources at the macro level and realize the sustainable use of land space.

Natural development scenario: This scenario was based on land-use data from 2020 and the land-use transition probability matrix from 2010 to 2020 (Table A2). According to the normal development of the city and without considering the constraints of various national land space planning, the Markov model was used to predict the scale of each land-use type in 2035, which was used as the basis for the input data of the FLUS model.

Cultivated land protection scenario: Cultivated land security is key to ensuring national food security and is an important cornerstone for the realization of high-quality life. In this scenario, the transfer of cultivated land to other land types was strictly limited to prevent large-scale occupation of cultivated land by urban expansion during economic development. Simultaneously, referring to relevant research [57], we reduced the transfer probability of cultivated land to construction land by 40% and to woodland by 30% on the basis of the natural development scenario.

Ecological protection scenario: The construction of an ecological civilization is an important means of alleviating the contradiction between the development of human activities and the sustainable utilization of natural resources. It is a major measure which ensures the service function of the ecosystem and meets the basic living needs of people. Based on the natural development scenario, we increased the protection of woodland, grassland, and waters, so the transfer probabilities of woodland and grassland to construction land and unused land were reduced by 80% and 100%, respectively, and the transfer probability of waters to construction land was reduced by 80%. This scenario increased the probability of transferring cultivated land to woodland by 30% and reduced the probability of transferring cultivated land to construction land by 30%.

#### 3.3.2. Land-Use Change Simulation Based on Markov–FLUS Model

(1)Land-use quantity demand simulation using the Markov model.

The Markov model mainly predicts the scale of land-use demand under different scenarios by adjusting the transfer probability matrix. Based on the data of land use in 2010 and 2020, which was converted to ASCLL format in ArcGIS software, we firstly reclassed these data by using the IDRISI software (Clark Labs, Clark University, Worcester, MA, USA). Then, with the help of the Markov module, we obtained the transfer area matrix. By using the CA–Markov model in IDRISI software, we finally obtained the amount of land use in the future. To reduce the error of the Markov model in predicting the scale of long-time-series land-use demand, after testing the simulation accuracy in 2020, we predicted the area of various land types in 2025, 2030, and 2035 and then used the prediction results in 2035 as the data basis for the FLUS model. The calculation formula is as follows:S_(*t*+1)_ = P*_ij_* S*_t_*(1)
where S*_t_* is the initial state of land-use type at *t*, S_(*t*+1)_ is the state of land-use type at (*t* + 1), and P*_ij_* is the transfer probability matrix of the land-use type.

(2)Calculation of suitability probabilities and neighborhood factors.

In the simulation process, the suitability probability was calculated using the occurrence probability module based on the ANN in the FLUS model. The calculation formula is as follows:(2)$$\mathrm{sp}\;(p,\;k,\;t)=\sum_{j}w_{i,j}\;\times \;\frac{1}{{1+e}^{{-\mathrm{net}}_{j}\left(p,t\right)}}$$
where sp (*p*, *k*, *t*) is the suitability probability of *k* land type output on the *p*—pixel at time *t*, w*_i_*_,*j*_ is the weight between the hidden layer and the output layer, and net*_j_* (*p*, *t*) is the signal of the input layer received by the *j*-th neuron on the *p*—pixel at time *t*.

In addition, neighborhood influence factors can reflect the interaction between different land types and land units within the neighborhood. In this study, a Moore 3 neighborhood window was selected to calculate this parameter. Its expression formula is as follows:(3)$$\Omega_{p,k}^{t}=\frac{\sum_{N\times N}{\mathrm{con}(c}_{p}^{t-1}=k)}{N\times N-1}{\times w}_{k}$$
where $\underset{N\times N}{\sum }{\mathrm{con}(c}_{p}^{t-1}=k)$ is the total number of pixels after the end of the last iteration (*t* − 1) of the neighborhood window land type *k* of *N*×*N*, *w_k_* is the neighborhood factor parameter of each land type, and $\Omega_{p,k}^{t}$ is the neighborhood influence factor of cell *p* at time *t*. The neighborhood parameter factor ranges from 0 to 1, and its value reflects the expansion capacity of the land type under the influence of various driving factors. The closer it is to 1, the stronger the expansion capacity of the land type. In this study, the neighborhood factor parameters were set with reference to previous relevant studies [58] and the degree of interference from human activities (Table 1).

(3)Adaptive inertia coefficient.

The adaptive inertia coefficient adjusts the land type in the next iteration according to the difference between the current land type and expected demand scale to achieve the expected goal. The formula used is as follows:(4)$$I_{k}^{t}=\{\begin{array}{ll} \;I_{k}^{t-1} & \;\;\;\;\;\;\mathrm{if}\;\;\;\left|D_{k}^{t-1}\right|\leq \left|D_{k}^{t-2}\right|\;\;\;\\ I_{k}^{t-1}\times \frac{D_{k}^{t-2}}{D_{k}^{t-1}} & \;\;\;\;\;\mathrm{if}\;\;\;D_{k}^{t-1}<D_{k}^{t-2}<0\\ I_{k}^{t-1}\times \frac{D_{k}^{t-2}}{D_{k}^{t-1}} & \;\;\;\;\;\mathrm{if}\;\;\;0<D_{k}^{t-2}<D_{k}^{t-1} \end{array}\;\;\;\;$$
where $I_{k}^{t}$ is the inertia coefficient of land type *k* at iteration time *t*, and $D_{k}^{t-1}$ is the area difference between the current land type and the expected target.

(4)Cost matrix.

The cost matrix represents the degree of difficulty in the mutual conversion between different land types. It is 0 when the land classes cannot be converted and 1 when the land classes can be converted. Combined with the actual land-use transformation, in the process of economic development, other land types are constantly converted to construction land, so the cost matrix of construction land is set to 0. However, the setting of other land types was judged according to the restricted state of transfer between land types in different scenarios [59]. Under the natural development scenario, except for construction land, other land types could be transferred to one other. Under the cultivated land protection scenario, cultivated land was set to not be transferred to other land types, and woodland and waters could not change each other. Under the ecological protection scenario, the cost matrix was arranged according to the principle of prohibiting the conversion from high-grade to low-grade land, resulting in a sequence, from high to low, of woodland, waters, grassland, and other land use. The cost matrix was set according to the aforementioned provisions (Table 2).

(5)Model accuracy test.

Based on the trend of land-use change from 2010 to 2020, after predicting the quantity and scale of land use in 2020 using the IDRISI software, this study used the FLUS model to simulate the land-use situation in that year (Figure A2). The simulation results were compared with real land use in 2020 and then the overall accuracy and kappa coefficient were used to verify the accuracy. The closer the overall accuracy and kappa coefficient are to 1, the more accurately a model can simulate the spatial change of regional land use. When the kappa coefficient is greater than 0.8, the model achieves better simulation accuracy with statistical significance [58]. In this study, the overall accuracy of the model was 0.952, and the kappa coefficient was 0.924, indicating that the FLUS model was suitable for simulating land-use change in the Guanzhong Plain urban agglomeration.

#### 3.3.3. Habitat Quality Assessment Based on the InVEST Model

This study evaluated the habitat quality of study area using the habitat quality module of InVEST model. Habitat refers to the space possessed by organisms that can provide resources and survival conditions [3]. The module fully reflects the threat of human activity on habitat quality. Under the influence of human activity, habitat quality is disturbed to varying degrees, resulting in a decline in biodiversity. The core of this module is determined to establish the relationship between threat sources and habitat quality. The evaluation mainly includes calculating the degree of interference of threat sources to the habitat, that is, the degree of habitat degradation, and calculating habitat quality in combination with the habitat suitability of various land types. The formula for calculating habitat degradation is as follows:(5)$$D_{xj}=\sum_{r=1}^{R}\sum_{y=1}^{Y_{r}}\left(\frac{\omega_{r}}{\sum_{r=1}^{R}\omega_{r}}\right)r_{y}i_{rxy}\beta_{x}S_{jr}$$
where *R* is the habitat stress factor, y is the number of grids in stress factor *r*, *w_r_* is the weight of different threat sources, *r_y_* is the intensity of stress factors, *i_rxy_* is the influence of the stress source in the grid of habitat, $\beta_{x}$ is the anti-interference level of the habitat, and *S_jr_* is the relative sensitivity of each habitat to different stress factors. The value of habitat degradation degree ranges from 0 and 1. The closer the value is to 1, the higher the degree of habitat degradation. The formula for calculating habitat quality is as follows:(6)$$Q_{xj}=H_{xj}\times \left[1-\left(\frac{D_{xj}^{2}}{D_{xj}^{2}+k^{2}}\right)\right]$$
where *Q_xj_* is the habitat quality of grid *x* in habitat type *j*, *H_xj_* is the habitat suitability of *j* land type, *D_xj_* is the degree of habitat degradation of *j* land type in grid *x*, and *k* is the semi-saturation constant. The value of habitat quality ranges from 0 and 1. 

The model mainly requires three parameters: the impact scope of threat sources, habitat suitability, and the relative sensitivity of each habitat to different threat sources. According to the current situation of the study area and relevant studies, cultivated land, construction land, national roads, provincial roads, and main railways were chosen as the threat sources. In addition, referring to the user manual of the investment model [15] and existing relevant studies [60,61], we assigned values to the threat factors and habitat sensitivity (Table 3 and Table 4).

## 4. Results

### 4.1. Characteristics of Land-Use Change in Guanzhong Plain Urban Agglomeration

In 2000, the most common land-use type of the Guanzhong Plain urban agglomeration was cultivated land at 46.28% of the total area, followed by woodland, grassland, and construction land at 21.48%, 26.82%, and 4.07%, respectively. With the continuous promotion of urban development activities and the orderly implementation of ecological projects, such as returning cultivated land to woodland (or grassland), the area of regional cultivated land decreased, accounting for 43.75% of the total area in 2020. Meanwhile, the areas of woodland, grassland, and construction land increased, reaching 21.74%, 27.41%, and 5.73%, respectively, in 2020. From 2000 to 2020, the land-use types of the Guanzhong Plain urban agglomeration underwent a dramatic transformation, mainly due to the transfer of cultivated land to other land-use types. The decrease in cultivated land area was 4422.75 km^2^, which was far higher than its increase (1704.91 km^2^). The reduced area was the main source of the growth of other land use, mainly flowing to grassland and construction land. Additionally, the decrease in woodland area (620.01 km^2^) was less than its increase (891.30 km^2^), being mainly transformed from cultivated land and grassland. Meanwhile, the increase in grassland area (2274.06 km^2^) was significantly higher than its decrease (1645.19 km^2^), of which 80.92% came from cultivated land. Construction land expanded rapidly through the continuous occupation of cultivated land, with the area increase reaching 2120.92 km^2^, of which 91.71% came from cultivated land (Table 5 and Figure 3). 

### 4.2. Multi-Scenario Simulation Results

Based on the land-use data in 2020, different scenarios were established to simulate the land-use change pattern of the Guanzhong Plain urban agglomeration in 2035 (Figure 4 and Figure 5).

(1)The natural development scenario allowed free conversion between various land types without considering the influence of national land space planning and development policies. In terms of quantity change, the area of cultivated land, woodlands, and grassland under this scenario showed a decreasing trend. Compared with 2020, the area of cultivated land, woodland, and grassland decreased by 1981.95 km^2^ (3.30%), 444.67 km^2^ (3.95%), and 456.76 km^2^ (1.57%), respectively, and the area of waters and unused land decreased slightly by 40.87 km^2^ (3.41%) and 5.43 km^2^ (3.71%), respectively. Only construction land showed an expansion trend, increasing significantly by 2934.4 km^2^ (54.82%). From the change in the land-use spatial pattern, construction land spread throughout the region, from Xi’an as the core to Xianyang City, Weinan City, Baoji City, Tianshui City, and other regions. Construction land in Linfen City and Yuncheng City also expanded rapidly based on the original distribution. In addition, cultivated land and ecological land were occupied and disturbed to varying degrees.(2)Under the cultivated land protection scenario, the expansion speed of construction land was restrained, and the transfer of cultivated land to other types of land was controlled. In terms of quantity change, the scale of cultivated land showed an increasing trend, increasing by 411.97 km^2^ (0.68%) from 2020, mainly because the transfer area of cultivated land to woodland, grassland, and construction land decreased. Grassland and unused land area increased slightly by 133.51 km^2^ (0.45%) and 15.05 km^2^ (10.27%), respectively. Compared with the natural development scenario, the expansion trend of construction land was lower, increasing by only 132.56 km^2^ (2.47%); meanwhile, the reduction in woodland area increased, decreasing by 672.32 km^2^ (5.98%), which showed that the transfer trend of cultivated land to woodland decreased. The waters area decreased slightly by 15.67 km^2^ (1.31%). From the perspective of the change of land-use spatial pattern, the expansion of construction land was mainly concentrated in Xi’an and its surrounding areas, followed by Linfen and Yuncheng. The cultivated land area mainly increased in the west of the Guanzhong Plain, such as in Pingliang City and Tianshui City, and its growth source was mainly woodland. This change was consistent with the direction of land development and utilization determined in the development plan of the Guanzhong Plain urban agglomeration.(3)Under the ecological protection scenario, to promote the ecological co-construction and environmental co-governance of the Guanzhong Plain urban agglomeration, various types of ecological land were protected, and the rate of transfer of ecological land to other land was controlled. In terms of quantity change, the cultivated land area decreased by 4176.49 km^2^, which was a larger decrease (6.96%) than in the other scenarios. The area of woodland decreased slightly by 1.91 km^2^ (0.02%), but the scale of woodland decrease was smaller than in the other scenarios. The unused land area decreased slightly (2.04%). Additionally, grassland and waters showed the most significant growth trends among the three scenarios, with their area increasing by 1839.08 km^2^ (6.31%) and 476.22 km^2^ (39.78), respectively. The area of construction land increased significantly, reaching 1870.53 km^2^, but, compared with the natural development scenario, its expansion rate was restrained, and its growth rate decreased from 54.82% to 34.94%. From the perspective of the change of land-use spatial pattern, construction land mainly expanded in Xi’an and Xianyang, with significant expansions also in Linfen and Yuncheng. The cultivated land reduction areas were mainly in the north and south of the Guanzhong Plain urban agglomeration and were mainly transferred to grassland and construction land. In other words, under the premise of ecological protection, it is necessary to meet the needs of various human activities so that various social and economic activities can operate normally.

### 4.3. Temporal and Spatial Variation of Habitat Quality

Referring to relevant studies [12,62], we used the natural breaks in ArcGIS software to divide the habitat quality value into four grades, poor (0–0.4), moderate (0.4–0.6), good (0.6–0.8), and excellent (0.8–1), and calculated the proportion of habitat quality grade area from 2000 to 2020 and in 2035 under various scenarios (Figure 6 and Figure 7).

(1)Temporal variations in habitat quality: The average habitat quality in 2000, 2010, and 2020 was 0.7188, 0.7141, and 0.7121, respectively, and the regional habitat quality decreased by 0.93% over 20 years. During this period, the habitat quality grade showed the characteristics of a high proportion of areas with moderate and excellent habitat quality and a low proportion of areas with poor and good habitat quality, while the area with poor habitat quality gradually increased from 3.54% in 2000 to 5.11% in 2020, mainly due to the shrinkage of areas with moderate and good habitat quality; meanwhile, the area with excellent habitat quality increased slightly (0.95%). In general, the habitat quality of the Guanzhong Plain urban agglomeration was good, and the overall level was high, but the habitat was in decline. Regarding the average habitat quality, the three scenarios were ranked as follows: ecological protection scenario > cultivated land protection scenario > natural development scenario. Under the natural development scenario, from 2020 to 2035, the average habitat quality continued to decline to 0.6921, the area with poor habitat quality increased to 7.87%, and the moderate-grade area decreased from 54.1% to 53.3%. This decline in the overall habitat quality of the region was mainly due to the continuous expansion of construction land to support economic development, encroaching on cultivated land, woodland, and grassland around the city. In 2035, the average habitat quality under the cultivated land protection scenario was 0.7082. Compared with the natural development scenario, habitat quality was improved, the proportion of low-value areas was lower, and the proportion of high-value areas was higher. The poor area was lower at 5.26%, the moderate-grade area was higher at 55.66%, and the excellent area was higher at 37.97%. The average habitat quality under the ecological protection scenario in 2035 was the highest of the three scenarios (0.7109). Although there was a gap in 2020, the poor area accounted for 6.86%, which was lower than the percentage under the natural development scenario, while the proportions of good-grade and excellent-grade areas were 1.55% and 40.23%, respectively, which were higher than in the other two scenarios, indicating that the ecological quality of this scenario was better.(2)Spatial pattern variations in habitat quality: During the study period, the habitat quality of the study area showed a spatial pattern of poor in the middle, moderate in the east and west, and good in the north and south. Areas with high habitat quality were mainly distributed in the south and north of the study area: the Qinba Mountains in the south, and the Loess Plateau in the north. Overall, the ecological environment was good. Areas with low habitat quality were mainly located in the middle of the urban agglomeration with Xi’an as the core. The economic development of this area was good, various development activities caused great damage to the habitat, and the urban space continued to extend to the surrounding higher habitat areas. From 2000 to 2020, the habitat quality of the study area continued to decline. With the disorderly expansion of urban construction land brought about by economic growth, the moderate-habitat-quality area gradually deteriorated. The habitat quality under the natural development scenario from 2020 to 2035 further decreased (Figure 6), the low-value area in the middle expanded, and the construction land in Linfen and Yuncheng in the northeast expanded. Although the habitat quality under the cultivated land protection scenario was reduced compared with the natural development scenario, the habitat quality was obviously higher. Cultivated land was protected while slowing the transformation of woodland and grassland, besides which, the expansion degree of construction land in the central region was significantly weakened, indicating that a cultivated land protection policy is conducive to slowing the decline of habitat quality. Compared with the previous two scenarios, the ecological protection scenario had the highest habitat quality, which enabled economic development in the central part of the Guanzhong Plain urban agglomeration and the protection of north–south habitats. Therefore, this scenario was conducive to economic, social, and ecological sustainability.

## 5. Discussion

### 5.1. Effects of Land-Use Change on Habitat Quality in Urban Agglomerations

Land-use change is an important driver of changes in habitat quality [2]. From 2000 to 2035, the habitat quality level of the Guanzhong Plain urban agglomeration was relatively high but generally showed a declining trend. The main reason was that, against the background of China’s rapid urbanization, the sharp expansion of urban and rural construction land and transportation land occupied a large amount of cultivated land [63], especially in the central area of the urban agglomeration with Xi’an as the core. With the implementation of the strategy for large-scale development of Western China [64], the new urbanization strategy [65] and “the Belt and Road” initiative [66], regional urbanization developed rapidly, and different land-use types were frequently transformed. Cultivated land, woodland, grassland, and waters were disturbed by the expansion of urban space, resulting in an obvious decline in habitat quality. However, the Loess Plateau in the north and the Qinling Mountains in the south of the urban agglomeration are the key implementation areas for China’s returning farmland to woodland (grassland) policy, the main functional area system, and the ecological civilization strategy. Since 2000, regional vegetation coverage has increased significantly, and habitat quality has improved significantly. Previous studies also found that, since 2000, the vegetation coverage in China’s Loess Plateau [67] and the Qinling Mountains [68] has increased significantly, and the overall habitat quality has improved.

### 5.2. Comparison of Land-Use Scenarios for Sustainable Development Goals

Considering the effects of natural conditions and human activities on land-use change, multi-scenario simulation of land use and the habitat quality of the Guanzhong Plain urban agglomeration was carried out by coupling the FLUS and InVEST models which accurately predicted the changes of vulnerable areas of the natural ecological environment around the city, making a significant contribution to the optimization of regional natural ecosystems, protection of human well-being, and protection of biodiversity [56]. Under the natural development scenario, regardless of policy planning constraints, construction land was expanded from the conversion of cultivated land, woodland, grassland, waters, and unused land to meet the land-use space required for economic growth. Blindly pursuing the rapid development of the urban economy sacrifices good ecological land originally held, resulting in the shrinkage of cultivated land, woodland, and grassland, which is not conducive to sustained economic growth and ecological security and stability. Under the cultivated land protection scenario, the transfer of cultivated land to other land types was limited, and the rate of construction land occupation of cultivated land was limited, which is conducive to protecting cultivated land, restraining the disorderly expansion of construction land and slowing the transfer of cultivated land to woodland. Under the ecological protection scenario, grassland and waters were fully protected, and the expansion scale of construction land was controlled, which is conducive to rational development activities on the premise of ecological protection. In general, the ecological protection scenario was the most conducive to regional urban development, cultivated land protection, and sustainable maintenance of ecological security and provides a basic scenario basis for building a sustainable geographic pattern in urban agglomerations and achieving sustainable development.

### 5.3. Policy Impact

Government policies have a far-reaching impact on the optimization of land-use structure and the improvement of ecological environment in urban agglomerations. The rapid urbanization and industrialization of the Guanzhong Plain urban agglomeration directly affects the transformation direction of land-use types, which has a profound impact on the quality of regional ecological environment [69]. With the adjustment of government policies, the habitat quality of the urban agglomeration has changed accordingly. The policy of returning cropland to forestland implemented in 2000 [70] promoted the transformation of cultivated land with a slope greater than 25° in the Loess Plateau in the north and Qinling Mountains in the south of the urban agglomeration into woodland and grassland, which has improved the overall ecological quality of the region. However, the central area of the urban agglomeration with Xi’an as the core was affected by the disorderly expansion of construction land during this period, and the habitat quality around the city decreased significantly. After 2010, the successive implementation of the main functional area planning and system [71] effectively guided the direction of land-use development and protection and strictly controlled the intensity and boundary of land-use development in the urban agglomeration. So, the change degree of land use in Guanzhong Plain urban agglomeration has taken a major turn. The intensity of land-use development and utilization has weakened; at the same time, the trend of habitat quality reduction has slowed down, which shows that national policies and systems have far-reaching significance in the development and utilization of land use in the Guanzhong Plain urban agglomeration. In 2014, the state established a new urbanization strategy with urban agglomeration as the core [65], pointing out the direction for the green and intensive development of urban agglomeration. In particular, the implementation of the Yellow River Basin ecological protection and high-quality development strategy [23] in 2018 optimized the development pattern of land use, which reduced the disorderly expansion of construction land and improved the habitat quality around the city. However, in recent years, the planning and construction of the Guanzhong Plain urban agglomeration and the greater Xi’an metropolitan area have still brought about a sharp increase in population and the expansion of urban production and living land, so the regional habitat quality is at risk of decline.

### 5.4. Limitations and Potentials

In this study, changes in land use and habitat quality in the Guanzhong Plain urban agglomeration were simulated by coupling the FLUS and InVEST models. The simulation results were highly accurate and effectively reflected the relationship between regional human activities and natural habitats. However, there were still limitations. First, although nine types of driving factor of land-use change, including natural and social aspects, required by the model were selected, the effects of climate, geology, and other factors were not considered. Second, although the InVEST model can spatially visualize habitat quality, it only considers the stress of each single threat factor on the habitat patch and ignores the synergistic effect of each threat factor on the habitat in the estimation process. Finally, for the influence scope and weight of threat sources, expert evaluation was only carried out by referring to relevant research, which is subjective to a certain extent. Therefore, in future research, it is necessary to appropriately increase the indicators of climate change, geology, landform, and the scope of nature reserves according to the regional reality. In addition, we will strengthen the localized processing and verification of the parameters of the investment model and carry out in-depth research on the driving mechanism and regulation strategy of regional habitat quality change to further improve the scientific and practical guiding value of the results.

## 6. Conclusions

With the continuous advancement of urbanization, land-use change has brought many problems, such as habitat fragmentation and loss of ecosystem services. Therefore, we simulated changes in the spatial pattern of land use and evaluated their impact on habitat quality based on a multi-scenario setting. Coupling the Markov model with the FLUS model helped to simulate and predict the changes in the spatial and temporal patterns of future land use in the study area under three different scenarios. Additionally, it was beneficial to use the InVEST model to evaluate change of habitat quality in different periods. The main conclusions are as follows:(1)Simulating the habitat quality of urban agglomerations based on land-use change is an important method for understanding and evaluating the complex coupling relationship between human activities and natural habitats. The Markov–FLUS model selected in this study showed excellent simulation results and was suitable for simulating land-use changes in the Guanzhong Plain urban agglomeration. The model test showed that the overall accuracy was 0.952, and the kappa coefficient was 0.924, indicating that the model had strong applicability for predicting future land-use change in the Guanzhong Plain urban agglomeration and effectively reflected the impact of regional human activities on the state and change in natural habitat;(2)From 2000 to 2020, the cultivated land area of the Guanzhong Plain urban agglomeration decreased by 5.48%, and the construction land area increased by 40.58%. Under the natural development scenario from 2020 to 2035, the growth in construction land area was the most significant, increasing by 54.82%. The cultivated land area only showed an upward trend under the cultivated land protection scenario, with an increase of 411.97 km^2^, while the increase in construction land area was the lowest at only 2.47%. Under the ecological protection scenario, the cultivated land area decreased by 6.96%, and the grassland and waters area showed a growth trend, increasing by 1839.08 and 476.22 km^2^, respectively;(3)From 2000 to 2020, the habitat quality of the Guanzhong Plain urban agglomeration gradually decreased from 0.7188 to 0.7121. The high-value areas of habitat quality were mainly distributed in the Qinling Mountains in the south of the Guanzhong Plain and the Loess Plateau in the north, while the low-value areas were mainly located in the middle of the urban agglomeration, with Xi’an as the core. In 2035, the order of average habitat quality under the three scenarios was: ecological protection scenario > cultivated land protection scenario > natural development scenario, showing that the ecological protection scenario was most conducive to building a sustainable geographical pattern in the urban agglomeration area, realizing the effective coordination of urban development, agricultural production, and ecological protection. It can also provide an important basis for the sustainable development of the urban agglomeration area.

## Figures and Tables

**Figure 1 ijerph-19-08703-f001:**
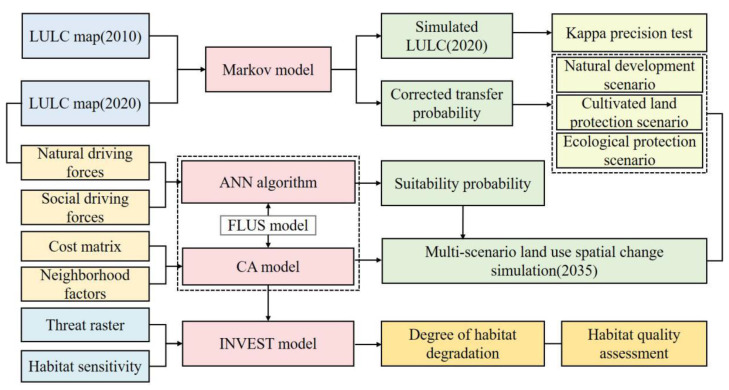
Research framework.

**Figure 2 ijerph-19-08703-f002:**
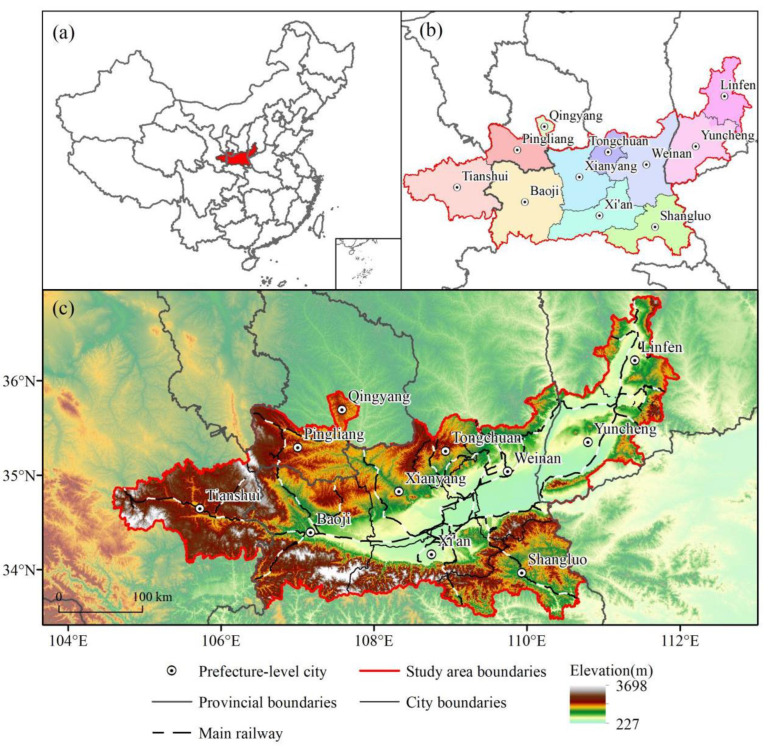
Overview of Guanzhong Plain urban agglomeration: (**a**) location in China; (**b**) administrative divisions; (**c**) elevation.

**Figure 3 ijerph-19-08703-f003:**
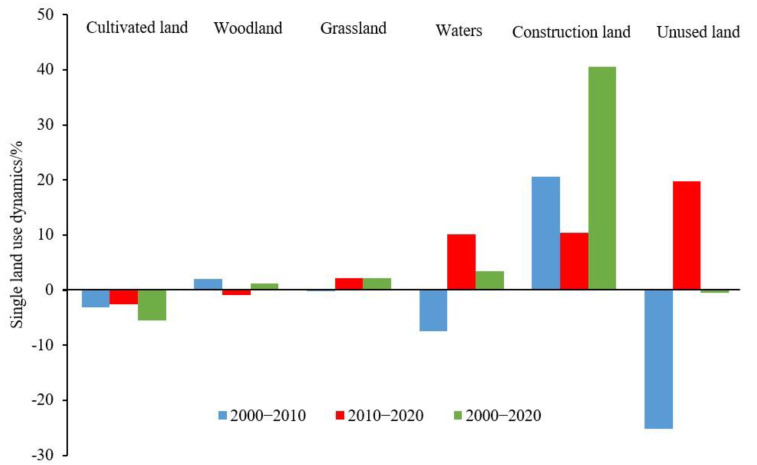
Dynamic degree change of single land use in Guanzhong Plain urban agglomeration from 2000 to 2020.

**Figure 4 ijerph-19-08703-f004:**
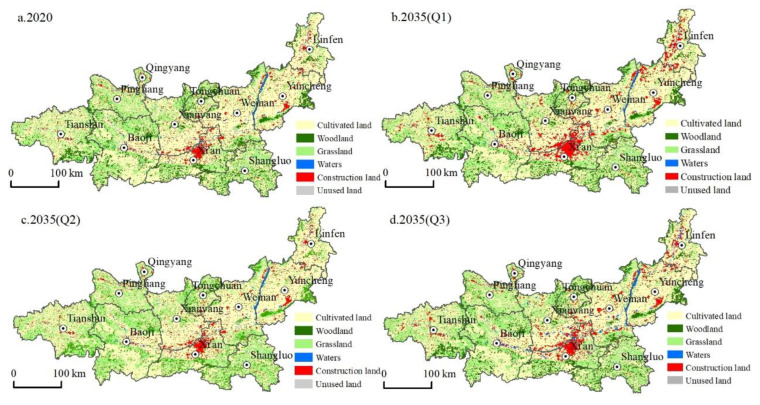
Multi-scenario land use simulation results.

**Figure 5 ijerph-19-08703-f005:**
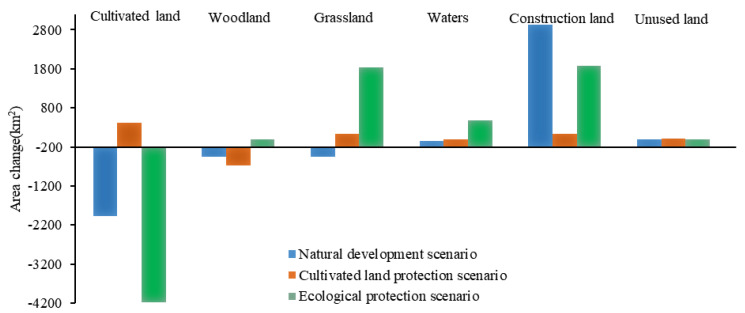
Comparison of land-use change scenarios from 2020 to 2035.

**Figure 6 ijerph-19-08703-f006:**
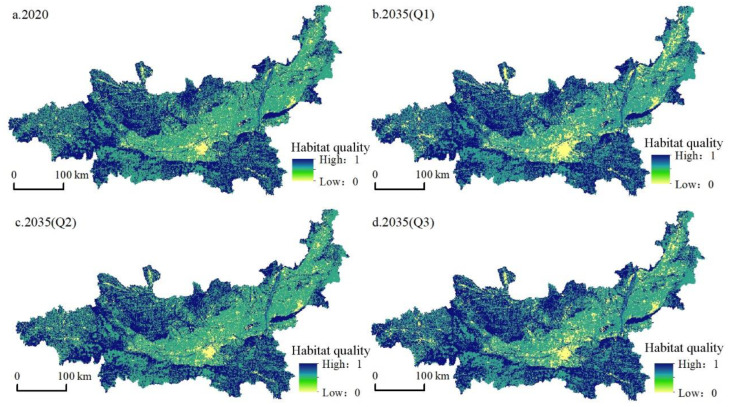
Spatial distribution of habitat quality in the Guanzhong Plain urban agglomeration under different scenarios in 2035.

**Figure 7 ijerph-19-08703-f007:**
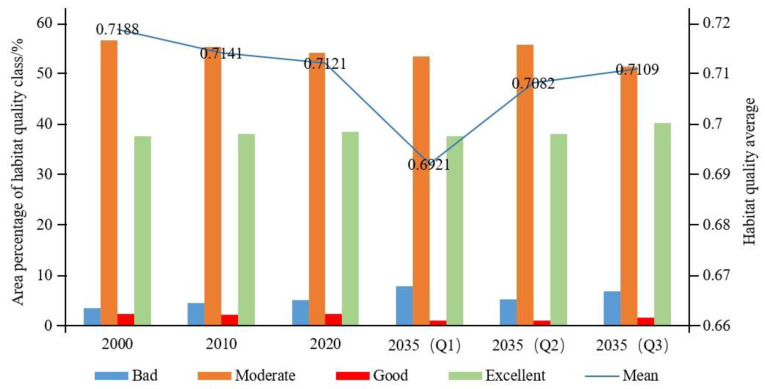
The ratio of habitat quality grade area in 2000, 2010, and 2020 and under each scenario in 2035.

**Table 1 ijerph-19-08703-t001:** Neighborhood factor parameters.

Land-Use Type	Construction Land	Unused Land	Waters	Grassland	Cultivated Land	Woodland
Neighborhood factor parameters	1	0.5	0.4	0.3	0.2	0.01

**Table 2 ijerph-19-08703-t002:** Multi-scenario cost matrix.

	Natural Development Scenario						Cultivated Land Protection Scenario						Ecological Protection Scenario					
	a	b	c	d	e	f	a	b	c	d	e	f	a	b	c	d	e	f
a ^1^	1	0	0	0	1	0	1	0	0	0	0	0	1	1	1	1	1	0
b	1	1	0	0	1	0	1	1	1	0	1	1	0	1	0	0	0	0
c	1	0	1	0	1	0	1	1	1	1	1	1	0	1	1	1	0	0
d	1	1	1	1	1	0	1	0	1	1	1	1	0	0	1	1	0	0
e	0	0	0	0	1	0	0	0	0	0	1	0	0	0	0	0	1	0
f	1	1	1	1	1	1	1	1	1	1	1	1	1	1	1	1	1	1

^1^ In the table, a, b, c, d, e, and f represent cultivated land, woodland, grassland, waters, construction land, and unused land, respectively. 1 indicates that the land class can be converted, and 0 indicates that the land class cannot be converted.

**Table 3 ijerph-19-08703-t003:** Threat sources and their coercion intensity.

Threat Source	Maximum Stress Distance (km)	Weight	Spatial Decay Type
Cultivated land	3	0.7	Linear decay
Construction land	10	1	Exponential decay
National roads	2	0.8	Linear decay
Provincial roads	2	0.8	Linear decay
Main railways	2	0.8	Linear decay

**Table 4 ijerph-19-08703-t004:** Habitat suitability and its relative sensitivity to different threat sources.

Land-Use Type	Habitat Suitability	Threat Source				
		Cultivated Land	Construction Land	National Roads	Provincial Roads	Main Railways
Cultivated land	0.6	0.3	1	0.4	0.4	0.3
Woodland	1	0.8	0.8	0.6	0.6	0.5
Grassland	1	0.7	0.7	0.4	0.3	0.2
Waters	0.8	0.5	0.9	0.5	0.4	0.4
Construction land	0	0	0	0	0	0
Unused land	0	0	0	0	0	0

**Table 5 ijerph-19-08703-t005:** Land-use transition matrix of Guanzhong Plain urban agglomeration from 2000 to 2020 (km^2^).

Land Types	Cultivated Land	Woodland	Grassland	Waters	Construction Land	Unused Land	Area Decrease
Cultivated land	45,189.71	437.62	1840.31	181.06	1945.30	18.46	4422.75
Woodland	183.38	22,410.22	367.73	10.41	49.10	9.39	620.01
Grassland	1063.22	434.71	27,107.45	38.64	92.82	15.80	1645.19
Waters	131.65	7.38	35.87	1043.12	29.92	3.16	207.98
Construction land	316.02	6.68	17.19	6.30	4026.26	0.32	346.51
Unused land	10.65	4.91	12.96	15.55	3.78	112.55	47.85
Area increase	1704.91	891.30	2274.06	251.96	2120.92	47.13	

## Data Availability

Data presented in this study are available from the corresponding authors upon request.

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
