# Peer review of "Multi-Scenario Simulation of Land Use and Habitat Quality in the Guanzhong Plain Urban Agglomeration, China"

_ijerph, 2022, doi:10.3390/ijerph19148703_

Round 1

Reviewer 1 Report

The research paper covers the theme of land use and habitat quality.

1. The paper should include more literature. Not enough relevant papers have been reviewed.

There is much  information such as geographic information that seems redundant, and makes paper difficult to read. Redundancy is an issue and should be addressed.

2. The introduction section is quite long. There is a need to indicate clearly what the value, originality and significance to the literature of the study is. 

3. Required to create a literature review section after introduction, and include new literature in the literature review section, and also implications for regional development , such as 

B.I., K, S.A. & A, S.P. (2015). Public Land Acquisition and Land Use Change Problems in Ogun State. International Journal of Management Science and Business Administration, 2(8), 34-41. and others

4. The research results are extensive. However they are not interpreted in the context of prior studies.The discussion and implications section is quite thin. 

5. Paper requires refinement, editing by the authors to focus on delivering the finidngs in a more convincing manner.

The research framework should be seperated from methodology section and enriched with literature upon which it has been developed

So the squence of section is 1. introduction 2. Literature review in which you could also include framework development and 3. Methodology

6. Proofreading is also needed

Reviewer 2 Report

Introduction: A very concise review is presented. However, there is a need for a clear presentation of the goal and aims at the end of this section.

Materials-Methods:

Study area: Since we are talking about land uses and habitat quality, I was expecting at this session to present shortly the main land uses of the area and a description of the habitat types and their current situation.

Methodology:

Fig. 2: What do you mean by Humanistic driving forces. Do you man-made driving forces?

Fig.2: Is ANN Model and CA model consists of FLUS model? This is not very clear on the diagram based on your text above fig. 2. Instead of saying model, you could say ANN algorithm? However, we are still missing from the diagram the FLUS model.

Lines 141-142 and lines 187-188. There is a confusion here, about the contribution of each model.

Results: At this section, the results could present only figures and the elaboration of them to feed the Discussion session. As such there will be a balance among the two sections (Results and Discussion). There are many conclusions at this section that could be included in the following ones (discussion, conclusions).

General comments:

It is a very interesting article, and very well written. However, the authors need to carefully look the comments and make the necessary revisions, in order for the article to be more clear, comprehensive and acceptable.

Reviewer 3 Report

This manuscript deals with one of the most significant and important topics for the definition of a supporting decision-making mechanism in urban planning. The evaluation of land-use change trends, the prediction with alternative scenarios and their evaluation by means of Habitat Quality performances.

Although this manuscript is without doubt interesting, there are some weaknesses that should be accurately checked and reviewed. 1) the introduction is somehow generic and without any in-depth preliminary assessment of what is the importance  and the computation techniques of land-use change analysis (see Verbourg and Veldkamp). 2) the methodology is only partially described, since there is no information about the land-use dataset, nor it isn't clear if this land use are auto-produced or is just processed by land-use change analysis. 3) the criteria to assign the HQ input values are hidden. 4) results are somehow really generic and the authors infer many aspects (fragmentation) that are not empirically measured by their analysis. 5) discussions are improper. This chapter just replicates the results without any sensible added value (policy implications). 6) Conclusions should recap the original research question, and then synthesize the methods and results. In the attached file you can find all my detailed comments.

Round 2

Reviewer 1 Report

the article has been improved to a certain extent. It can further be improved regarding the same points, polished and proofread.

Reviewer 3 Report

The authors worked extensively in the suggested direction and the manuscript gained more quality. 

I just ask the authors to carefully re-read thought the text... and correct punctuation, repetitions, or typing mistakes (there are many here and there).

Then, to my view, the manuscript can be accepted for publication
